# LATENT DIFFUSION WITH LLMS FOR REASONING

## ABSTRACT

Despite the widespread adoption of large language models with hundreds of billions of parameters, these models still struggle on complex reasoning benchmarks. In this paper, we argue that the autoregressive nature of current language models are not suited for reasoning due to fundamental limitations, and that reasoning requires slow accumulation of knowledge through time. We show that combining latent diffusion models with an encoder-decoder transformer architecture provides a scalable way to address some of the fundamental shortcomings posed by autoregressive models. Diffusion models can arrive at predictions through many forward passes in latent space, and their reasoning is not handicapped by the order of the tokens in the dataset. Through our experiments, we show that latent diffusion language models is a feasible approach towards scalable language models that have general complex reasoning abilities.

## 1 INTRODUCTION

In recent years, autoregressive large language models (LLMs) have become the de-facto for natural language generation (Team et al., 2024; Radford et al., 2019). The excellent scalability of transformers combined with the availability of large datasets has led to many practical applications where language models elicit impressive emergent capabilities. However, even the biggest corporate LLMs still struggle with complex reasoning benchmarks (Sawada et al., 2023). Prior work has shown that LLMs are limited by their autoregressive nature because the FLOPs used to generate each token is constant regardless of the difficulty of the token (Bachmann & Nagarajan, 2024). Additionally, the model has to generate tokens in the order of the dataset it is trained on and therefore cannot solve easier subproblems first (Bachmann & Nagarajan, 2024). The model will not explicitly generate easier reasoning chains first unless explicitly fine-tuned to do so on a subset of tasks.

Numerous approaches have tried to tackle this problem such as chain-of-thought (CoT) prompting (Wei et al., 2022), enhancing model reasoning with longer context (Dai et al., 2019), or encoding recurrence into transformers (Hutchins et al., 2022; Bulatov et al., 2022). These approaches, however, are not a general solution to these shortcomings because pretraining datasets are rarely CoT prompted, and compute allocated to each token is still constant. This is a fundamental limitation when solving math problems. For example, not all tokens have equal difficulty, and often times the answer to easier subproblems lead to better answers for harder subproblems (e.g. geometry, algebra). Even though recurrent models can perform many forward passes in latent space, prior work has not been able to scale efficiently due to its memory requirements, and it has been observed that long unrolls lead to exploding or vanishing gradients (Vicol et al., 2021).

In this paper, we propose to combine latent diffusion models (LDMs) with encoder-decoder transformers in an attempt to solve the mentioned shortcomings that are posed by autoregressive LLMs. In contrast to traditional LLMs, LDMs generate a latent vector by iteratively denoising Gaussian noise throughout many timesteps, which intuitively makes it more suitable for tasks that require extrapolating many facts over long horizons. Prior work has shown that text diffusion models elicit self-correction abilities, where reasoning steps can be generated non sequentially and that the model learns to correct its wrong answers from easier (and correct) reasoning steps (Ye et al., 2024). LDMs perform reasoning in latent space which is semantically richer than discrete tokens. For a specific task, the required semantics might not be representable by its token embeddings (e.g. spatial reasoning). LDMs also do not suffer from memory requirements and instabilities encountered by back-

propagation through time (where it is a common practice to set max timesteps to 1000 or 4000). This is due to the fact that gradients are not propagated through the same parameters multiple times (Ho et al., 2020; Nichol & Dhariwal, 2021) which makes it an appealing candidate to solve the aforementioned shortcomings. It has also been shown that latent diffusion text models outperform discrete diffusion text models, strengthening our claim that operating in latent space yields improvements over operating with discrete tokens (He et al., 2022; Lovelace et al., 2024b).

We summarize the benefits of combining LDMs with encoder-decoder language models for complex reasoning task as follows:

1. It can do reasoning in semantic space and does not rely on discrete tokens where the accumulation of knowledge per forward pass only amounts to that particular generated token.

2. It can perform reasoning non-sequentially regardless of the order of the tokens in the training data. Throughout denoising steps, LDMs elicit self-correction where correct reasoning steps lead to corrections on harder reasoning steps.

3. It does not run into memory bottlenecks and instabilities that are encountered by recurrent transformers as we scale to larger unroll lengths because gradients are not propagated through the same parameters multiple times.

## 2 RELATED WORK / PRELIMINARIES

Generative pretrained transformers (GPTs) have significantly transformed natural language processing demonstrating exceptional scalability and achieving state-of-the-art performance on a variety of downstream tasks, including translation, summarization, and instruction following (Achiam et al., 2023). Meanwhile, image generation also had a renaissance powered by LDMs (Yang et al., 2023). By iteratively denoising an image distribution from Gaussian noise, diffusion models have been able to outperform generative adversarial networks on image generation benchmarks. Continuing research on LDMs have also found that these models generate more diverse image samples, and techniques such as Min-SNR-$\gamma$ (Hang et al., 2023) and progressive distillation (Salimans & Ho, 2022) improved the efficiency of the training and inference such that LDMs can now generate high quality images and videos at a fraction of the cost (Rombach et al., 2022). Since this paper combines diffusion with autoregressive encoder-decoder language models, we briefly review the literature on the reasoning abilities of LLMs and some basic concepts to understand diffusion models.

### 2.1 DENOISING DIFFUSION PROBABILISTIC MODELS

DDPMs (Ho et al., 2020) are a class of diffusion models that iteratively construct an image from random Gaussian noise. We define $x_0$ as the original image which is slowly corrupted into random Gaussian noise iteratively. The forward process, which converts the original image into a corrupted image (by adding Gaussian noise), can be formulated as $x_t = \sqrt{\bar{\alpha}_t} x_0 + \sqrt{1 - \bar{\alpha}_t} \epsilon$ where we sample noise $\epsilon \sim \mathcal{N}(0, I)$, and $\bar{\alpha}_t$ is a noise scheduling hyperparameter that controls how noise is applied on different timesteps. In order to learn the reverse process to reconstruct the target image $x_0$, a model $\theta$ is learned to predict the noise at each timestep, which is optimized by minimizing a simple mean squared error loss between $\hat{\epsilon} = \epsilon_\theta(x_t, t)$, the estimated noise of time $t$, and $\epsilon$: $\mathcal{L}_{\text{simple}}(\theta) = \|\epsilon_\theta(x_t, t) - \epsilon\|_2^2$. During each iteration of inference, random Gaussian noise can then be turned into an image according to a target data distribution by iteratively removing $\epsilon_\theta(x_t, t)$. For each sampling step, the denoised image for the next step is given by $x_{t-1} = \frac{1}{\sqrt{\alpha_t}} \left( x_t - \frac{1 - \alpha_t}{\sqrt{1 - \bar{\alpha}_t}} \epsilon_\theta(x_t, t) \right) + \sigma_t z$ where we sample noise $z \sim \mathcal{N}(0, I)$, $\sigma_t$ is the standard deviation, and $\bar{\alpha}_t = \prod_{s=1}^{t} \alpha_s$.

### 2.2 SCALABLE DIFFUSION MODELS WITH TRANSFORMERS

Diffusion transformers (DiT) is a variant of transformer that has been modified to incorporate conditioning information for diffusion (Peebles & Xie, 2023). Conditional image generation can be formulated as the probability of an image $x$ given information such as a class label $c$, $p_\theta(x|c)$, where $c$ is an additional information (such as the class of an image). The DiT architecture consists of adaLN-zero blocks (Peebles & Xie, 2023) which incorporate conditioning information by regressing dimension-wise over the scale and shift parameters used in adaptive layer normalization

(adaLN) from the sum of the embedding vectors of the current timestep $t$ and class $c$. In addition to adaLN, DiT also regresses dimension-wise scaling parameters that are added prior to any residual connections. They further initialize all multilayer perceptron to output the zero-vector for $\alpha$ since this initializes the residual block to an identity block (by adding zero to the residual connections) which leads to faster convergence empirically. DiT has shown to outperform traditional U-Net as backbone for diffusion due to its remarkable scaling properties.

### 2.3 DIFFUSION-OF-THOUGHT

Due to the difficulty of reasoning tasks, LLMs perform poorly when they are tasked to directly output an answer to a difficult reasoning problem. Therefore, CoT is a technique to improve LLM accuracy by fine-tuning it to output a reasoning chain before the final answer (Wei et al., 2022). This increases LLMs performance on hard reasoning benchmarks because the model can generate easier reasoning first that can aid it in finding the final answer. Diffusion-of-thought (DoT) attempts to take it a step further by having a discrete diffusion model diffuse CoT tokens. The authors found out that DoT elicits self-correction abilities which is in contrast to traditional LLMs Huang & Chang (2022). Our work attempts to take it a step further by augmenting it with LLMs so that it can get the best of both worlds (efficient pretraining and strong reasoning abilities).

### 2.4 STEP-DPO

Mathematical reasoning is recognized as a long-chain reasoning ability in LLMs (Lai et al., 2024). Previous work has tried to tackle this by applying Direct Preference Optimization (DPO) (Rafailov et al., 2024) to the reasoning chain with the correct answer but with limited success. Step-DPO addresses this issue by applying DPO to each reasoning step, and curate a dataset that contains pairwise preference data generated by the model itself, which has been shown to improve training compared to GPT-4 generated data and human labeled data Lai et al. (2024). With our proposed model architecture, we show that diffusion can be an add-on to step-DPO.

### 2.5 INTEGRATING MONTE CARLO TREE SEARCH FOR LLM REASONING

Contuining work on LLM reasoning has turned to Monte Carlo tree search (MTCS) to create a self-improving loop without the addition of annotated data (Tian et al., 2024). Since outputs from MCTS are usually in much better quality, the gap ensures that LLM can continue to self-improve. Results show that this method improves performance by as much as 30% for GSM8K and MATH benchmarks Tian et al. (2024). Our proposed method can also be added on to the MTCS-based reasoning approaches.

## 3 PROPOSED METHOD

In this paper, we propose to merge the encoder-decoder language model with LDMs in an attempt to enhance reasoning in natural language processing. We use pretrained encoder-decoder LLMs as our base model since these LLMs already contain high-level semantics that have been learned from large corpus of text. Particularly, we use BART (Lewis et al., 2019) extensively throughout our experiments to obtain its encoder representations. In contrast to next sequence prediction, the decoder is fine-tuned to generate the original sequence given the encoder representation.

The main training consists of two stages. First, we fine-tune the decoder and an autoencoder such that the variable length encoder representation can be compressed to a fixed length latent, which can then be decoded back to its original token sequence. This improves reliability and efficiency because diffusion models are more compute efficient at training smaller dimensional latent variables and input tokens inherently have different lengths. Second, we train a diffusion model such that the diffusion transformer denoises the target sequence compressed latent conditioned on the input sequence compressed latent. Reasoning is achieved by iteratively constructing the target latent through many forward passes in the diffusion model.

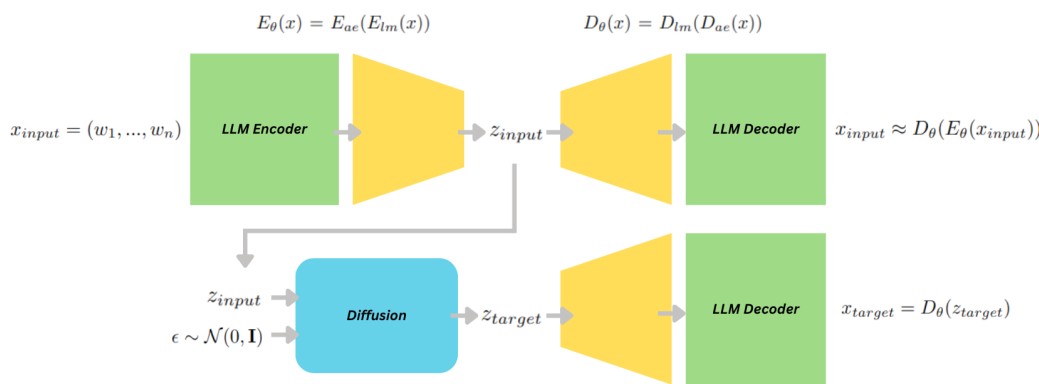

Figure 1: Overview of our proposed architecture to improve reasoning.

During inference, the input sequence is fed into the encoder and autoencoder to obtain the compressed input latent which is then used to condition the diffusion transformer to generate a compressed target latent. This compressed representation is then passed through the autoencoder and decoder to generate the predicted target sequence.

The following sections describe in more detail the architecture of the latent models and diffusion models and how they are trained.

### 3.1 LATENT MODELS FINE-TUNING

Formally, we define a sequence of tokens as $x = (w_1, ..., w_n)$ which are sampled from a dataset to get its inputs and corresponding targets $(x_{input}, x_{target}) \sim \mathcal{D}$. Then, we aim to learn a language autoencoder $\theta$ such that $x$ can be reconstructed by passing through an encoder $E_\theta$ and decoder $D_\theta$; that is, $x \approx D_\theta(E_\theta(x))$. In our setting, $E_\theta(x) = E_{ae}(E_{lm}(x))$, and $D_\theta(x) = D_{lm}(D_{ae}(x))$ where both the encoder and decoder are composed by an autoencoder denoted by $E_{ae}$ and $D_{ae}$ and a pretrained BART encoder and decoder denoted by $E_{lm}$ and $D_{lm}$, respectively. Since changes in the dimensionality of the latent representation can lead to drastic changes in final performance (He et al., 2022; Nichol & Dhariwal, 2021), we compress the encoder representation to a fixed latent space with length $l_{ae} = 16$ and dimension $d_{ae} = 256$. The autoencoder architecture consists of only cross-attention transformer blocks where first block queries are learned from learnable hyper-parameters of the target dimensions, and key and values are learned from the encoder representation or compressed representations. We did not ablate over the autoencoder design choice. Our goal is to have the compressed latents contain both low level features and high level semantics instead of simply compressing the token sequences, so we freeze $E_{lm}$ during all training stages because high level semantics are obtained from BART since they are not retrained to overfit on simply compressing the data.

### 3.2 LATENT DIFFUSION

The second stage simply consists of learning a diffusion model in the latent space learned by the autoencoder. The compressed latents with length $l_{ae} = 16$ and dimension $d_{ae} = 256$ are then projected up to dimension $d_{proj}$ which is then reshaped into length $l_{diffusion}$ and a fixed dimension $d_{diffusion} = 768$. Since DiT scales with decreasing patch size (or increasing sequence length), we ablate sequence length for DiT to determine whether the same scaling law holds for latent text diffusion. We follow the standard DDPM approach to train $x_0$ and train the variance $\Sigma_\theta$ with the full loss $\mathcal{L}(\theta) = -\log p_\theta(x_0|x_1) + \sum_t \mathcal{D}_{KL}(q^*(x_{t-1}|x_t, x_0)\|p_\theta(x_{t-1}|x_t))$. In preliminary experiments, we observed that predicting $x_0$ instead of $\epsilon_\theta$ was crucial to generate coherent text, and that pretrained encoder-decoder transformers are not sensitive to small pertubations of encoder representations. We use a cosine schedule with the max timestep as $T = 1000$ since higher $T$ improves the

log-likelihood of the generated samples (Nichol & Dhariwal, 2021), and that we can always sample more efficiently using different samplers from the literature that trade off sample quality.

## 3.3 Improvements

In addition to using diffusion to predict the target tokens, we could alternatively concatenate the input token sequences or representations to the decoder input to allow the decoder to do additional computations before outputting tokens to improve Perplexity. If we add noise to the encoder representation during training, it learns to differentiate between noise and signal from representations, teaching it that the diffusion output contains useful semantics but are not always reliable. Alternatively, we could also use encoders trained with contrastive learning to improve the quality of the latent representations. This allows the architecture to retain GPT performance while being able to solve additional reasoning tasks with diffusion output. If the latent representations from diffusion is unreliable, then the model defaults to autoregressive inference (Lovelace et al., 2024a). We opt to only give diffusion output to the decoder since the performance improvements would also depend on the decoder which would not reflect the capabilities of diffusion.

## 3.4 Comparison Against State-of-The-Art

Previous related work shows that discrete diffusion models have reasoning potential but lack the training efficiency to rival traditional LLMs (Gulrajani & Hashimoto, 2024). By combining encoder-decoder transformer with diffusion, we can leverage the best of both worlds: training efficiency and enhanced reasoning. Since representation is learned using traditional LLMs, the diffusion model is able to directly utilize high-level semantics without the inefficiencies of training diffusion. Additionally, if the decoder takes in input tokens alongside output from diffusion, it can selectively utilize signals from the diffused latent representation for complex reasoning tasks while discarding noise if the output is not useful (such as when optimizing for Perplexity instead of BertScore). This work does not overlap most prior work on reasoning, making it a suitable add-on to the state-of-the-art reasoning techniques such as Monte Carlo tree search (Xie et al., 2024) or graph of thought (Besta et al., 2024).

# 4 Results

Throughout our experiments, we study the potential benefits and the scalability of our approach to augment encoder-decoder LLMs with diffusion to enhance reasoning. Specifically, we test this approach against tasks that aim to measure arithmetic reasoning and spatial reasoning. Additionally, we ablate different architecture variants, diffusion sequence length, and model layers to determine the best architecture for scaling. We then summarize our findings to provide insights on how to scale this hybrid architecture.

## 4.1 Arithmetic Reasoning

To analyze the performance of the proposed hybrid architecture with downstream math tasks, we create single digit addition problem sets where 3-5 single digit numbers are added together. CoT reasoning chains are provided as the target where the model is trained to iteratively add the first two digits. The model is required to output the first token as an answer along with its subsequent reasoning. Table 1 presents comparison between the performance of latent diffusion and fine-tuned BART for arithmetic tasks.

Table 1: Single digit additions

| Architecture | Accuracy↑ |
| --- | --- |
| Latent Diffusion (T=500) | 97.2 |
| Latent Diffusion (T=1000) | 96.7 |
| Latent Diffusion (T=4000) | 97.3 |
| BART (First token as answer) | 1.3 |
| BART (Last token as answer) | 0.3 |

We further study the proposed hybrid model's performance by testing out different arithmetic tasks that mirrors the arithmetic experiments done for GPT-3 (Brown, 2020). We observe that latent diffusion performs remarkably well for its given model size. We acknowledge that this might not be a fair comparison because GPT-3 is not fine-tuned for arithmetic tasks but it should still reflect model capacity and scaling law.

Table 2: Double digit additions.

| Architecture | Accuracy↑ |
|---|---|
| Latent Diffusion (T=1000, 140M) | 87.2 |
| BART (fine-tuned) | 0.0 |
| GPT-3 (400M) | 5.0 |
| GPT-3 (13B) | 57.0 |
| GPT-3 (175B) | 99.0 |

Table 3: Single digit three operations.

| Architecture | Accuracy↑ |
|---|---|
| Latent Diffusion (T=1000, 140M) | 100.0 |
| BART (fine-tuned) | 11.8 |
| GPT-3 (400M) | 2.5 |
| GPT-3 (13B) | 10.5 |
| GPT-3 (175B) | 21.0 |

Given the same number of training iterations, our findings show that the proposed architecture learns various arithmetic tasks while BART fails completely. The reason is that predicting the first token as answer leads to worse performance for the encoder-decoder because it is unable to self-correct after giving an incorrect answer and have to give subsequent reasoning for the wrong answer (OOD). This is an advantage of diffusion because pretraining data scraped from the internet are rarely well behaved (ordered from easy tokens to hard tokens).

## 4.2 MOCK SPATIAL REASONING

To study the benefits of latent diffusion augmented LLMs against conventional LLMs, we create a mock spatial reasoning problem where four numbers are presented as input and the model is tasked with coming up with the answer as the first token and subsequent reasoning. The reasoning consists of rotations of $up \rightarrow down \rightarrow left \rightarrow right \rightarrow up$. Initially, we start with $up$, then rotate $n$ times where $n$ is the first number, and each subsequent number reverses the direction of the rotation. For example, given input *1 3*, the output should be **left**. Specifically, the output sequence is **left** *up down up right* where **left** is the final answer. We first start with *up*, then rotate one time to *down*, then reverse direction and rotate three times to *left*.

The problem consists of easy reasoning chains which is required for computing the first token. However, coming up with the first token directly is nearly impossible. In practice, reasoning might not be representable by tokens but the model could still rely on high-level semantics learned by the BART model's encoder-decoder.

Table 4: Mock spatial reasoning for the rotation task.

| Architecture | Accuracy↑ |
|---|---|
| Latent Diffusion (T=500) | 90.4 |
| Latent Diffusion (T=1000) | 92.3 |
| Latent Diffusion (T=4000) | 89.5 |
| BART (fine-tuned) | 0.0 |

The Encoder-Decoder performs as good as the random baseline throughout training, and predicted the end of text token as the answer near the end of training. We show from the results that latent

diffusion does not have to rely on the order of the dataset and can do easy reasoning chains to extrapolate harder answers. Many hard reasoning problems in the real world are impractical to be represented by CoT tokens, therefore, doing reasoning in latent space could be a promising alternative. Augmenting latent diffusion also has an additional benefit when there is many repetition in the reasoning chain. For example, if reasoning requires many multiplication arithmetics, diffusion is able to reuse its layers to compute many repetitive multiplications throughout many timesteps, whereas autoregressive models can only use the same layer once to produce a token.

### 4.3 ABLATIONS

We first ablate different architectures to incorporate input text conditioning since text latents could have different properties compared to image class labels (Table 5 presents the results). Throughout experiments, we found that in-context conditioning has minimal compute overhead, while having negligible difference on BertScore, hence we adopt in-context conditioning for most of the ablations. We use the Common Crawl (C4) dataset for all of the ablation experiments since it includes a variety of different sequences from most domains.

*In-Context* · We concatenate the noised target representation sequence with the input representation sequence. The output is split into two sequences, where the first one is the model output, and the second one is the predicted variance.

*Cross-Attention* · An additional cross-attention module is added after self-attention for each DiT block to incorporate input text conditioning.

*AdaLN-Zero* · Input text conditioning information is incorporated by adding it to the timestep representation which is fed into the AdaLN-Zero block similar to the DiT architecture for class labels.

We further observe improved performance with increasing depth. However, we observe a weak negative correlation between both metrics and loss with increasing diffusion sequence length which is in contrast to image diffusion transformers. This suggests that further architecture improvements can be made or scaling should be done through increasing layers and not sequence length. We further observe that BertScore does not always correlate with Perplexity, which leads us to hypothesize that the loss function that optimizes representation could sacrifice coherence for semantic similarity. Hence we use BertScore as the main metric for determining performance since it also correlates better with loss, whereas Perplexity has very high variance and depends significantly on the target sequence length and architecture. Images are known to be more parallelizable since there are more independent patches whereas text data are more interdependent.

To further study the effects of high-level semantics learned by the encoder-decoder architecture, we compare the performance of BART-base (140M parameters) and BART-large (406M parameters) to determine whether the improved quality of both low- and high-level representations also carries over after augmenting with LDMs of the same size. The results show that diffusing better representations from pretrained weights improves BertScore.

We highlight that for this experiment, instead of compressing the last representation of the encoder, we compress the concatenation of the first and last representation of the encoder. This is due to the observation that the decoder did not provide an accurate reconstruction of the original text from only the last encoder representation of BART-large.

We've found that generation length decreases with longer training times from preliminary experiments which led to uncomparable BertScores since longer sequences have higher BertScores on average and that shorter sequences have lower Perplexity on average. One hypothesis is that the diffusion model only denoises signals from earlier tokens since they have lower variance (e.g. the next token is easier to predict than the 16th token), leading to later positions denoised as paddings. Since experiments are trained with different hyperparameters for different learned generation lengths, metrics cannot be compared between different experiments. Further research on how to

Table 5: Model, architecture, sequence length, and model depth ablation studies

| Architecture | Layers | Sequence Length | BertScore↑ |
|---|---|---|---|
| In-Context | 6 | 16 | 70.0 |
| In-Context | 12 | 16 | 70.2 |
| In-Context | 24 | 16 | 70.6 |
| In-Context | 6 | 16 | 70.0 |
| In-Context | 6 | 32 | 70.2 |
| In-Context | 6 | 64 | 69.6 |
| Cross-Attention | 6 | 16 | 70.2 |
| Cross-Attention | 12 | 16 | 70.4 |
| Cross-Attention | 24 | 16 | 70.4 |
| Cross-Attention | 6 | 16 | 70.2 |
| Cross-Attention | 6 | 32 | 70.0 |
| Cross-Attention | 6 | 64 | 68.1 |
| AdaLN-Zero | 6 | 16 | 69.7 |
| AdaLN-Zero | 12 | 16 | 69.8 |
| AdaLN-Zero | 24 | 16 | 70.1 |
| AdaLN-Zero | 6 | 16 | 69.7 |
| AdaLN-Zero | 6 | 32 | 69.0 |
| AdaLN-Zero | 6 | 64 | 70.1 |

Table 6: Comparison between BART-base and BART-large.

| Model | BertScore↑ |
|---|---|
| BART-base | 67.64 |
| BART-large | 69.80 |

better evaluate these variable length diffusion models will be required to improve the reliability of current metrics.

## 5 IMPLEMENTATION DETAILS

Throughout our experiments, we adopt classifer-free guidance to improve sample quality at the expense of sample diversity. We also use Min-SNR-$\gamma$ because it improves the training efficiency.

### 5.1 CLASSIFIER-FREE GUIDANCE

Classifier-free guidance is widely known to improve sample quality. (Ho & Salimans, 2022; Nichol et al., 2021). By jointly training the unconditional $p_\theta(x)$ and conditional $p_\theta(x|c)$ model for a specific class $c$, we can sample using a linear combination of the score estimates. This is relatively straight-forward to implement by randomly setting $p_\theta(x|c = \varnothing)$ during training. Classifier-free guidance can be used to encourage the sampling procedure such that $log\ p(c|x)$ is high and tradeoff between sample quality and diversity.

### 5.2 MIN-SNR-$\gamma$

Min-SNR-$\gamma$ (Hang et al., 2023) improves the training efficiency by weighing each loss term as $w_t = min\{\text{SNR}(t), \gamma\}$ where $t$ is the timestep and $\gamma$ is a hyperparameter. By taking into account the signal-to-noise ratio (SNR), min-SNR-$\gamma$ is better able to traverse the loss landscape by weighing conflicting gradients between earlier and later diffusion steps. Furthermore, Min-SNR-$\gamma$ takes the minimum between SNR($t$) and $\gamma$ to avoid the model focusing too much on small noise levels. All training run uses $\gamma = 5$ as our weighing strategy.

## 6 DISCUSSION

It has been known that diffusion language models yield better diversity when generating text. We show from our work that augmenting latent diffusion with language models outperforms autoregressive models for certain reasoning cases. A notable limitation of diffusion is that it is relatively inefficient to train compared to conventional language models. One hypothesis is that there is a combinatorial explosion as more tokens are diffused at once, hence the gradients are not as well-behaved (noisy gradient landscapes). As research on diffusion continues, we should expect that it will play a more prominent role in natural language processing to address some of our current limitations.

An exciting research direction would be to utilize the proposed architecture for idea generation while implementing the ideas with the same architecture specialized for coding. This could be a feasible approach towards artificial general intelligence (AGI) because it has more diverse ideas, and it can directly implement the programs by reasoning about the structure of the code in latent space beforehand. This could initiate recursive self-improvement, leading to increasingly automated deep learning research. However, due to its inefficiencies and other potential obstacles, it remains uncertain how far we can practically scale such architectures with current hardware and algorithms.

## 7 CONCLUSION

Reasoning involves extrapolating across many facts over extended horizons. In this paper, we demonstrated that augmenting latent diffusion with encoder-decoder architecture outperforms autoregressive language models in scenarios where tokens have different levels of difficulty (more reasoning required), and that adhering strictly to the sequential order of the dataset is not beneficial for accuracy. We propose that this architecture offers a promising approach for solving real-world reasoning tasks by operating in latent space. To our knowledge, this is the first work exploring the augmentation of latent diffusion for reasoning. As research on diffusion models continue to narrow the gap with autoregressive models, we are optimistic that this new architecture can achieve better reasoning with further scale and advancements.

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

## A  TRAINING HYPERPARAMETERS

### A.1  ABLATING ARCHITECTURE

| Hyperparameters | In-Context | Cross-Attention | AdaLN-Zero |
|---|---|---|---|
| Diffusion sequence length | 16 (each input) | 32 | 32 |
| Depth | | 12 | |
| Batch size | | 128 | |
| Sequence Length | | 64 | |
| Latents sequence length | | 8 | |
| Latents dim | | 256 | |
| Hidden size | | 768 | |
| Number of heads | | 12 | |
| Total timesteps (T) | | 1000 | |
| Learning rate | | 1e-4 | |
| Iterations | | 200k | |
| Floating Point | | float32 | |

### A.2  ABLATING DEPTH

| Hyperparameters | 6 Layers | 12 Layers | 24 Layers |
|---|---|---|---|
| Layers | 6 | 12 | 24 |
| Architecture | | In Context | |
| Diffusion sequence length | | 16 | |
| Batch size | | 128 | |
| Sequence Length | | 128 | |
| Latents sequence length | | 16 | |
| Latents dim | | 256 | |
| Hidden size | | 768 | |
| Number of heads | | 12 | |
| Total timesteps (T) | | 1000 | |
| Learning rate | | 1e-4 | |
| Iterations | | 200k | |
| Floating Point | | float16 | |

### A.3 ABLATING DIFFUSION SEQUENCE LENGTH

| Hyperparameters | In Context 16 | In Context 32 | In Context 64 |
|---|---|---|---|
| Diffusion sequence length | 16 | 32 | 64 |
| Batch size | | 128 | |
| Sequence Length | | 128 | |
| Latents sequence length | | 16 | |
| Latents dim | | 256 | |
| Hidden size | | 768 | |
| Number of heads | | 12 | |
| Layers | | 6 | |
| Total timesteps (T) | | 1000 | |
| Learning rate | | 1e-4 | |
| Iterations | | 200k | |
| Floating Point | | float16 | |

### A.4 ABLATING ENCODER REPRESENTATIONS

| Hyperparameters | BART-base | T5-large |
|---|---|---|
| Diffusion sequence length | | 64 |
| Batch size | | 128 |
| Sequence Length | | 128 |
| Latents sequence length | | 16 |
| Latents dim | | 256 |
| Hidden size | | 768 |
| Number of heads | | 12 |
| Layers | | 18 |
| Total timesteps (T) | | 1000 |
| Learning rate | | 1e-4 |
| Iterations | | 200k |
| Floating Point | | bfloat16 |

### A.5 MOCK SPATIAL REASONING FOR THE ROTATION TASK.

| Hyperparameters | LD (T=500) | LD (T=1000) | LD (T=4000) | Encoder-Decoder |
|---|---|---|---|---|
| Diffusion sequence length | | 16 | | - |
| Sequence Length | | 128 | | - |
| Latents sequence length | | 16 | | - |
| Latents dim | | 256 | | - |
| Hidden size | | 768 | | - |
| Number of heads | | 12 | | - |
| Layers | | 24 | | - |
| Total timesteps (T) | 500 | 1000 | 4000 | - |
| Pretrained Encoder-Decoder | BART-base | BART-base | BART-base | BART-base |
| Batch size | 128 | 128 | 128 | 128 |
| Learning rate | 1e-4 | 1e-4 | 1e-4 | 1e-4 |
| Iterations | 500k | 500k | 500k | 500k |
| Floating Point | bfloat16 | bfloat16 | bfloat16 | bfloat16 |

## A.6 BIG TRAINING RUN

| Hyperparameters | Main Run |
|---|---|
| Architecture | Cross-Attention |
| Diffusion sequence length | 32 |
| Sequence Length | 128 |
| Latents sequence length | 16 |
| Latents dim | 256 |
| Hidden size | 768 |
| Number of heads | 12 |
| Layers | 24 |
| Total timesteps (T) | 1000 |
| Pretrained Encoder-Decoder | BART-base |
| Batch size | 128 |
| Learning rate | 1e-4 |
| Floating Point | bfloat16 |

## A.7 MULTISTEP ADDITION

| Hyperparameters | LD (T=500) | LD (T=1000) | LD (T=4000) | Encoder-Decoder |
|---|---|---|---|---|
| Diffusion sequence length | | 16 | | - |
| Sequence Length | | 128 | | - |
| Latents sequence length | | 16 | | - |
| Latents dim | | 256 | | - |
| Hidden size | | 768 | | - |
| Number of heads | | 12 | | - |
| Layers | | 24 | | - |
| Total timesteps (T) | 500 | 1000 | 4000 | - |
| Pretrained Encoder-Decoder | BART-base | BART-base | BART-base | BART-base |
| Batch size | 128 | 128 | 128 | 128 |
| Learning rate | 1e-4 | 1e-4 | 1e-4 | 1e-4 |
| Iterations | 500k | 500k | 500k | 500k |
| Floating Point | bfloat16 | bfloat16 | bfloat16 | bfloat16 |

