# OpenReview forum: "Latent Diffusion with LLMs for Reasoning"
_ICLR.cc/2025/Conference — Submitted to ICLR 2025_

### Official Review · Reviewer_Enpx · 2024-10-20

**Soundness:** 2
**Presentation:** 2
**Contribution:** 2
**Rating:** 3
**Confidence:** 3

**Summary:**

The paper proposes combining latent diffusion with an encoder-decoder transformer architecture for reasoning tasks. The authors argue that such an approach address the weaknesses of autoregressive language models for reasoning tasks: namely, that such models must allocate a constant amount of compute per token and are unable to break complex tasks into easier tasks without explicit CoT finetuning. The proposed methodology involves using a fine-tuned BART encoder/decoder transformer model as well as a trained autoencoder model to map input/target tokens into a latent space; a diffusion model is then trained to map input latents to target latents. The authors demonstrate the effectiveness of their latent diffusion approach on two tasks (addition and mock spatial reasoning), comparing performance against a fine-tuned BART model. They also perform various ablation experiments using BERTscore as the performance metric.

**Strengths:**

- The paper addresses important limitations inherent to autoregressive language models on reasoning tasks.
- The original approach of using the pre-trained BART encoder/decoder to generate latents then applying diffusion to said latents avoids the costly training associated with previous diffusion language model approaches.

**Weaknesses:**

- The paper is unclear on how its proposed method differs from and/or builds upon previous diffusion language model approaches, e.g. Ye et al. 2024. Is the primary novelty the use of the fine-tuned BART encoder/decoder model, instead of training this component alongside the diffusion model?
- The empirical results also do not compare the proposed methodology with any previous diffusion language model approaches; the only comparisons are against fine-tuned BART and GPT-3 (which, as the authors point out, is not a fair comparison because GPT-3 is not fine-tuned for the evaluation tasks.)
- I found the writing to be unclear and vague in many places.
	- For example, the description of the autoencoder architecture lacks precision and is difficult to parse: "The autoencoder architecture consists of only cross-attention transformer blocks where first block queries are learned from learnable hyperparameters of the target dimensions, and key and values are learned from the encoder representation or compressed representations". Perhaps adding some equations or a diagram would make this understandable.
	- There are few details on the architecture of the diffusion model itself.
	- It would help the reader's understanding to have a few explicit examples of input and target sequences for both the evaluation tasks. (There is one example provided for the mock spatial reasoning task, which is good.)
	- I had difficulty precisely understanding the ablation experiments run. Adding more details could help clarify them.
- Having the final answer be the first target token is a somewhat contrived setup that adversarially targets the weakness of the autoregressive baseline model. This doesn't seem like the most natural or fair way to compare their performance.
	- Although, for the single digit addition task, the accuracy for "last token as answer" is provided (Table 1). In this case, BART somehow does even worse, which is very surprising. Although it's not a direct comparison, consider Shen et al. "Positional Description Matters for Transformers Arithmetic", which is able to obtain highly nontrivial performance on n-digit multiplication (arguably a harder task) using an autoregressive language model (GPT2-small, 117M parameters) without CoT.
- The proposed method requires supervised fine-tuning on target CoT sequences for well-defined synthetic tasks. It is unclear to me how it can be applied to more realistic settings.
- The fixed latent size is a major limitation -- roughly speaking, there is a fixed cap on the amount of intermediate reasoning steps the model can store in the latent, and it is unable to adapt to tasks of differing difficulty.

**Questions:**

- The latent diffusion model is listed is having 140M parameters (Table 2, 3). Is this just for the diffusion component? How many parameters are in the autoencoder model ($E_{ae},D_{ae}$)? Is the fine-tuned BART (I guess also 140M parameters) not included?
- There isn't a clear correlation between accuracy and timesteps $T$ in Tables 1 and 4. Why doesn't increasing timesteps increase accuracy?

---

> ### Author Response · Authors · 2024-11-21
>
> Thanks for reviewing the paper. Here are the clarifications we’ve made.
>
> - No, the BART-base model contains 140M parameters while the BART-large model contains 406M parameters independently, the parameter count in the autoencoder model is not as relevant because it is able to compress the text near perfectly, and changing the parameter count has negligible effect on compression quality. The fine-tuned BART models have the same amount of parameters.
>
> - It could be due to interfering gradients for training with larger timesteps; it has been known that gradients from different timesteps could interfere with each other. However, we did not try to find out why this is the case for that particular experiment.
>
> - The primary novelty is to test out diffusion augmented language models on reasoning benchmarks that do not require world knowledge since world knowledge would depend on the pretrained weights.
>
> - The autoencoder takes in a learnable embedding as input, then uses cross attention to incorporate BART representation such that the output has the same dimension as the learnable embedding.
>
> - The architecture of the diffusion model is just a Diffusion Transformer (DiT).
>
> - The architecture ablation tests three different kinds of layers to incorporate conditioning along with depth ablation.
>
> - We can use the same training scheme for pretraining data and not just fine tuning data.

---

> > ### Comment · Reviewer_Enpx · 2024-11-26
> >
> > Thank you for the response. It addressed a few of my questions.
> >
> > Overall, the paper still has significant weaknesses:
> > - Unclear writing, which has not yet been revised. (Reviewer 7307 gives a more thorough list of places where the writing is lacking).
> > - I'm still skeptical that a fine-tuned autoregressive model is unable to do well at the simple tasks used in the paper. The fact that it does even worse when it's only required to give its answer in the last token is surprising.
> > - The fixed-size latent seems like it would be a significant weakness for more complicated tasks.
> >
> > Thus I will be leaving my score as is for now.

---

### Official Review · Reviewer_NSA6 · 2024-10-28

**Soundness:** 2
**Presentation:** 2
**Contribution:** 3
**Rating:** 5
**Confidence:** 3

**Summary:**

The authors note that LLMs suffer on tasks that require a lot of forward planning/complex reasoning due to the same finite compute budget per token. They suggest combining diffusion models with an encoder-decoder architecture that uses denoising to generate the latent vector to be decoded. This means that the model can do many reasoning steps in latent space, rather than being forced to emit a token, and then read the token back into latent space.

They take a pre-trained encoder $E_{lm}$ / decoder $D_{lm}$ model (BERT), and train an autoencoder with encoder $E_{ae}$ and decoder $D_{ae}$ to compress the variable length encoding down to a fixed size latent vector $z_{input}$ such that the original encoding can be recovered from the compressed representation $x \approx D_{lm}(D_{ae}(z_{input}))$,
with $z_{input} = E_{ae}(E_{lm}(x))$. They then train a diffusion model to generate the compressed latent $z_{target}$ that is fed into the decoder $D_{ae}$ of the autoencoder, and then the pretrained decoder $D_{lm}$ to predict the target sequence $x_{target}$. $E_{lm}$ is frozen during training.

The authors measure the performance of the diffusion model on arithmetic tasks and some toy spatial reasoning tasks, and find it to be competitively performant with LLMs of a much higher parameter count.

**Strengths:**

Strengths:
* A new architecture that combines existing neural network architectures in a novel way
* The architecture at a high level was clearly explained.
* All hyperparameters for training were included in the appendix.

**Weaknesses:**

Weaknesses:
* Most open-source pre-trained transformers are of the decoder-only variant. This work would be most impressive if it could be adapted atop any open-source LLM, and selectively introduced to give the model the ability to "think" more/plan ahead for problems that require it.
* The results for the paper seemed weak to me: the performance of the model was only tried on toy tasks that specifically require forward planning.
* I would have liked to see at least in an appendix a full description of precisely what each task was for which the results were measured.
* Other more standard benchmakrs (e.g. MMLU) should have been included. If the model would be expected to perform poorly on these due to the simplistic model (BERT) it would build upon, and/or MMLU would be unsuitable as it's mostly a task for measuring recall, then that's fine, but it should have been mentioned.
* I think benchmarks from the original work on BERT would have been good to include, and compare how this performs as compared to this version.
* Some sample outputs from the model would have been nice to include?
* A plot of performance v.s. number of denoising steps would have been nice to include. The ability to adjust the amount of compute used during inference time is nice, so a comparison against a comparable LLM around the BERT-era would have been good (i.e. for at least a compute budget of X, our architecture outperforms LLM XYZ, and the performance plateaus at ... given x1, x5, x10, x50, ... amount of computing during inference time/number of denoising steps.)
* Sections 5.1 and 5.2 are very brief, and don't sufficiently motivate the use of these techniques.
* The internals of the autoencoder and diffusion architecture are very vague, and can't be inferred from the paper.

**Questions:**

Questions: (Annotated with Line number Num:)
* 126-143: Both Step-DPO and MCTS methods can add on to LLMs, but these methods are not used anywhere in the paper,
and the suggested architecture is not tested with them. I would remove these paragraphs and just mentions that your architecture
is amenable to fine-tuning methods as LLMs are.
* 190: Writing $x_{input} = (w_1, \ldots, w_n)$ seems strange. Why not just $x_{input} = (x_1, \ldots, x_n)$?
* 213: It's not entirely clear what the DDPM loss is training for here
	* Add a citation for the DDPM training method
	* What is $p_\theta$, what is $q^*$?
	* How were $l_{ae}$ and $d_{ae}$ chosen? Did you sweep over these hyperparameters in any way?
* 301-308: I did not follow this description of the "Mock Spatial Reasoning" task, it made no sense to me. Maybe include an image with it?
* Why are the benchmarked provided suitable? I've seen multi-digit addition before, but not this task
* 335: Need to define architecture ablation.

---

> ### Author Response · Authors · 2024-11-21
>
> Thanks for reviewing the paper. Here are the clarifications we’ve made.
>
> - Some of the suggestions will definitely be included in the next iteration of the paper.
>
> - The DDPM loss is training to approximate the target representation using MSE. In this case, p_\theta is the predicted distribution while, q* is the target distribution from the data.
>
> - We did not sweep over the l_{ae} and d_{ae}
>
> - For the Mock Spatial Reasoning, the first token is the answer, while the rest of the tokens are subsequent reasoning, we start with the initial token and change direction with each digit, each time rotating n times according to the digit (the list wraps around at each end).
>
> - We chose this benchmark because it doesn’t require world knowledge from the model to validate its reasoning, and they are also what was used to measure arithmetic reasoning in GPT-3.
>
> - For each different ablation, we test different layer architectures, in context, cross attention, and adaLN-zero, and also test different model depth.

---

> > ### Comment · Reviewer_NSA6 · 2024-11-26
> >
> > We thank the authors for their response.
> > We feel our concerns were adequately addressed, and do not have any further questions or concerns.
> > Unless there is disagreement on your end, we will leave our score as-is.

---

### Official Review · Reviewer_BGPR · 2024-10-31

**Soundness:** 1
**Presentation:** 1
**Contribution:** 1
**Rating:** 1
**Confidence:** 5

**Summary:**

The authors of the paper attempt to study whether diffusion on the latent space of a text auto-encoder can solve simple reasoning tasks such as single digit additions. The authors argue that regular transformers cannot adaptively change the amount of computation allocated to each token while other approaches such as diffusion might. However, there are no experiments on dynamic adaptation of the computational resources based on the task difficulty for their proposed solution.

**Strengths:**

- The premise of this paper, investigating whether text diffusion models offer stronger capabilities than autoregressive models is interesting and important.

**Weaknesses:**

- Many sentences and paragraphs are unclear. I have given my best to collect most examples but I might have forgotten some of them (examples listed below)
- Many claims are insufficiently supported by evidence (either from other papers or experiments). Similarly, I listed multiple examples below.
- The experiments are very limited and too simple (single digit addition, and a "spatial reasoning" task). It is expected that a model a model fine-tuned on the task performs better than zero-shot.
- The comparison between the author's method and GPT-3 seems unfair since they evaluate models trained for the simple addition task (in-distribution/training distribution evaluation) against GPT-3 zero-shot capabilities.
- The approach is surprising. The authors claim interest in "long chain reasoning" tasks such as mathematics (line 128), and tasks that require a variable amount of computation per token. At the same time, their approach relies on an auto-encoder that represents arbitrary-length inputs into a fixed-size tensor. As such, the approach does not look promising for long-horizon tasks.
- Section 3.3 is unexpected. Before discussing the experiments, the authors describe many possible improvement on their method.
- When I started reading this paper, I thought that the authors were planning to improve the reasoning capabilities of language models using a variable amount of *inference* computation per token. In section 3.4, the authors argue that they do not consider discrete diffusion models because of their slow *training* computation cost. However, a larger training cost could be acceptable if it produces a model with an adaptive computation mechanism. I would say those two questions (training vs inference costs) are separate, and it would be acceptable to study a model that is slow to train in their context.
- In section 3.4, authors argue that discrete diffusion models "have reasoning potential", and cite [4](https://arxiv.org/abs/2305.18619). However, there are no reasoning experiments in [4](https://arxiv.org/abs/2305.18619)


### Examples of unclear sentences and paragraphs
#### Abstract
- "In this paper, we argue that the autoregressive nature of current language models are not suited for reasoning due to fundamental limitations, and that reasoning requires slow accumulation of knowledge through time". What do you mean by time? Are you talking about the sequence length? If yes, then I would argue that causal transformers are slowly creating a representation based on their input.
- "...their reasoning is not handicapped by the order of the tokens in the dataset". Do you mean the order of tokens in the *sequence*? The order in the dataset should not have a direct influence if the data is shuffled correctly.
#### Rest of the paper
- Lines 39-41: " and often times the answer to easier subproblems lead to better answers for harder subproblems (e.g. geometry, algebra).": this is unclear to me. Can you provide me with 2 examples from geometry or algebra, where the answer to easier subproblems lead to better answers for harder subproblems?
- Lines 41-42: "Even though recurrent models can perform many forward passes in latent space,": are you talking about recurrent neural networks? I am not aware of work on recurrent neural networks operating in a latent space, beyond processing embedded tokens, similarly to a regular, decoder only transformer. Additionally, Vanilla RNNs also spend a constant amount of computation per input token, just as transformers
- Lines 52-53: "The required semantics might be representable by its token embeddings (e.g. spatial reasoning)": I dont understand this sentence, and the example neither, you need to be more explicit.
- 158-160: "Second, we train a diffusion model such that the diffusion transformer denoises the target sequence compressed latent conditioned on the input sequence compressed latent.": unclear
- 212: "We follow the standard DDPM approach to train $x_0$ and train the variance $\Sigma_\theta$" The best results in DDPM were obtained by predicting the noise, not $x_0$, hence you are not following the standard DDPM parameterization.

### Examples of insufficiently supported claims
Lines 47-49: " LDMs generate a latent vector by iteratively denoising Gaussian noise throughout many timesteps, which intuitvely makes it more suitable for tasks that require ex- trapolating many facts over long horizons.": I don't see why iterative denoising is "intuitively" more suitable for "extrapolating many facts". I don't understand the relationship between Gaussian noise and extrapolation.
- Lines 51-52: "LDMs perform reasoning in latent space which is semantically richer than discrete tokens." What evidence do you have that the latent space is richer? Are you saying this because you assume that continuous space are always richer? Since you are training your own auto-encoder, you need to demonstrate that the space you train is indeed richer.
- 57-59: you are claiming that latent diffusion models are stronger than discrete diffusion models. I would argue the opposite. The recent breakthroughs in diffusion language modeling used a discrete diffusion formalism. See [1](https://arxiv.org/abs/2406.07524), [2](https://arxiv.org/abs/2310.16834), [3](https://arxiv.org/abs/2406.04329) for example.
- Line 78-79: "diffusion models have been able to outperform generative adversarial networks on image generation benchmarks": yes, but you need a citation there
- Lines 129-130: "Previous work has tried to tackle... but with limited success": citation needed
- Lines 156-158: "This improves the reliability and efficiency, because diffusion models are more compute efficient a training smaller dimensional latent variables and input tokens inherently have different lengths": evidence needed
- Lines 217-218: " we can always sample more efficiently using different samplers from the literature that trade off sample quality.": citation needed

**Questions:**

## Suggestions
- If you are interested in adaptive computation (e.g. spending more inference resources per token based on the problem difficulty), the following two papers would be relevant to your research, both of which are purely based on transformers and do not require diffusion (simpler solution)
    - Mixture-of-Depths: Dynamically allocating compute in transformer-based language models (https://arxiv.org/abs/2404.02258)
    - Universal Transformers (https://arxiv.org/abs/1807.03819)
- Choice of tasks: it would be good to compare with tasks that are widely accepted by the community instead of new tasks such as the "spatial reasoning one". For example, you could have a look at the PicoCLVR tasks (https://fleuret.org/cgi-bin/gitweb/gitweb.cgi?p=picoclvr.git;a=tree) for some toy spatial reasoning tasks.
- A major source of improvement lies in the experimental setup. As stated in the "strength" section, studying whether diffusion models are stronger than autoregressive models is important. However, to convince the reader, you need to evaluate your models on more complicated tasks that are recognized by the community.
- As stated earlier, I do not believe that a fixed-size embedding is suited for variable-length problems. This is something you could investigate in further effort, as if you decide to continue using fixed-size embedding, you need to convince the reader that it does not limit the reasoning abilities of the model.
- Be careful of claims without evidence. For example, in the abstract, you state that your solution is "scalable". As such, the reader would expect that you scale your solution to larger scales (larger model, training set, training/inference FLOPs), and show that the performance continues to improve as you grow the model or resources.


I hope my review will help you refine your work.

---

> ### Author Response · Authors · 2024-11-21
>
> Thanks for the suggestions and comments. We have made some clarifications regarding your review.
>
> - The idea was not to do experiments on dynamic computation but to see whether there is an improvement in reasoning accuracy if the model could reuse the entire representation for the next forward pass through many iterations as opposed to selecting a token in the vocabulary for CoT reasoning.
>
> - In preliminary experiments, predicting x_0 leads to better results than e_\theta, this is inline with previous work [1].
>
> - We do not have enough compute to run the experiments on more complex tasks. Furthermore, the model would be limited by how well BART is trained and wouldn’t reach competitive performance against more recent LLMs.
>
> - As the time of writing, latent diffusion has outperformed past discrete diffusion models on language generation [1].
>
>
> [1] Justin Lovelace, Varsha Kishore, Chao Wan, Eliot Shekhtman, and Kilian Q Weinberger. Latent diffusion for language generation. Advances in Neural Information Processing Systems, 36, 2024b.

---

> ### Comment · Reviewer_BGPR · 2024-11-21
>
> Thank you for your reply. I respectfully believe that there are points from my review that remain unanswered. I would appreciate if you highlighted to which question you are answering before your answer. For example, you could use the following format:
> > Question of the reviewer
> - Answer from the authors
>
> ### Answer to the authors
> > We do not have enough compute to run the experiments on more complex tasks. Furthermore, the model would be limited by how well BART is trained and wouldn’t reach competitive performance against more recent LLMs.
> - I understand the constraint on computational resources. Because of the simplicity of your tasks, you should be able to tackle them with small models. [This work](https://arxiv.org/pdf/2307.03381) studies small autoregressive models for addition tasks for example.
>
> > As the time of writing, latent diffusion has outperformed past discrete diffusion models on language generation [1].
> - I respectfully disagree with the authors. I would say that the referenced work compares with relatively old/weak discrete diffusion models. In the past year, multiple work on discrete diffusion have been published, and they all seem to beat continuous diffusion for language modeling. See for example [SEDD](https://arxiv.org/pdf/2310.16834), [MDLM](https://arxiv.org/pdf/2406.07524) and [MD4](https://arxiv.org/pdf/2406.04329).

---

### Official Review · Reviewer_fxd5 · 2024-11-04

**Soundness:** 2
**Presentation:** 1
**Contribution:** 2
**Rating:** 3
**Confidence:** 4

**Summary:**

The proposed paper attempts to bridge the 'reasoning' gap in LLMs by training a text-diffusion model (continuous latent space), motivated by limitations of existing autoregressive LLM capabilities to allow for a) self correction abilities, b) adaptive compute based on token prediction difficulty. The work introduces a BART based conditioning mechanism to essentially compress text into the latent space using the fine-tuned BART, and then conduct continuous diffusion over the compressed latent. Thus, at training time, the autoencoder is first trained to allow for accurate compression followed by diffusion on the decoder transformer. During inference time, the input is compressed via the autoencoder, used to decode the target latent, which goes through the same input autoencoder to produce the final target sequence. The authors present result on two limited textual reasoning benchmarks, showing somewhat superior performance compared to auto-regressive counterparts.

**Strengths:**

The premise of non-autoregressive language modelling is promising, especially taking inspiration from the success of LDM in the image generation space. I also like the mock reasoning task as it represents a fundamental reasoning capability without having to test on an LLMs 'world' knowledge from large scale pretraining, similar in spirit to the ARC AGI benchmark. [1]. Further, leveraging diffusion denoising can be thought of as an interesting way to leverage test time compute.
[1] https://arcprize.org/arc

**Weaknesses:**

While an interesting and promising premise, there seem to be some unsupported claims (where it would be great to have some qualitative / quantitative preliminary analysis supporting the claim). Further, the results demonstrated in the paper are very limited and do not touch upon textual generation capabilities besides reasoning for supporting the claims. Finally, the authors do not seem to make a rigorous comparison with existing notable work on text diffusion (pointed in next section). The paper outlines drawbacks w.r.t existing LLMs but it is unclear if the proposed method would scale to potentially address these issues (e.g. reasoning performance on canonical benchmarks like GDM8k / MMLU). For example, it is claimed that "...are not a general solution to these shortcomings because pretraining datasets are rarely CoT prompted, and compute allocated to each token is still constant. This is a fundamental limitation when solving math problems." However, the reason CoT has been successful is due to the presence of STEM-like informal reasoning data present in the pre-training corpus, at scale. More concrete questions surrounding these have been outlined in the next section, where a discussion would be beneficial. In general, I would recommend the authors to narrow the scope of the proposal and not attempt reasoning as a whole, but rather conduct rigorous intuitive analysis to support the use of NAR (non autoregressive generation) for both text and reasoning.

**Questions:**

I would love to hear from the authors regarding the following points:
- How would you estimate or demonstrate the ability your approach model on pure text generation, say even on canonical textual benchmarks like SQuAD? It is hard to understand if such a method can be scaled to solve challenging reasoning tasks in the informal, textual domain. Before reasoning, how do the text generation capabilities compare? Or, is that not in scope at all? (it seems that would be surpising)

- It seems that this approach is very similar to SeqDiffuSeq[1] which is also based on BART based denoising, except for choosing to compress the target into the latent as well. In general, how would you compare your approach to simply training a 'embedding' head over the compressed latent (as compared to having a separate BART model for the auto-encoding)?

- Could you demonstrate some qualitative examples of sequences obtained from different models (e.g. on the 3 digit addition benchmark) to understand why the proposed approach can 'self correct' presumably through the iterative denoising, while autoregressive methods like GPT3 may not (thus being at 20% performance only)? The paper would benefit from a few example decodings.

- Can one leverage strong, pretrained embeddings to conduct diffusion without the BART involved at all? For example Gecko [2] embeddings, InstructOR[3] embeddings or others from the MTEB[4] leaderboard.

- Intuitively, where exactly is the gain in reasoning coming from, say compared to GPT? This point is related to the need for qualitative examples to understand why the approach is useful.

[1] Yuan, Hongyi, et al. "Seqdiffuseq: Text diffusion with encoder-decoder transformers." arXiv preprint arXiv:2212.10325 (2022).

[2] Lee, Jinhyuk, et al. "Gecko: Versatile text embeddings distilled from large language models." arXiv preprint arXiv:2403.20327 (2024).

[3] Su, Hongjin, et al. "One embedder, any task: Instruction-finetuned text embeddings." arXiv preprint arXiv:2212.09741 (2022).

[4] Muennighoff, Niklas, et al. "MTEB: Massive text embedding benchmark." arXiv preprint arXiv:2210.07316 (2022).

---

> ### Author Response · Authors · 2024-11-21
>
> Thanks for the suggestions and comments. Here are the clarifications according to the questions.
>
> - For this architecture, it is not the most performant at handling textual data, we do not have enough compute to create a completely fluent model based purely on this architecture. However, since the results show that it outperforms BART in certain fine-tuning cases, we could augment the input with the diffusion output as described in the improvements section. In this way, we could keep the text generation capability for decoder only GPT while leveraging the benefits of diffusion to improve performance on those tasks. We do not test the architecture mentioned in the improvement because the results would mostly reflect the decoder performance.
>
> - According to the paper “Diffusion Guided Language Modeling”, it does perform better compared to using the encoder-decoder architecture provided that the embeddings are well trained. However, this increases the cost of adding diffusion to the architecture because the decoder would have to be trained from scratch.
>
> - The text throughout the denoising is similar to discrete diffusion models, initially it starts out as a random sequence of tokens, then gradually corrects to noise to produce a coherent output. The token for answer would be wrong initially, then throughout each iteration be replaced by a more sensible token until the token is eventually correct.
>
> - Yes, it has been tested in other related work from the 2nd question.
>
> - The gains come from the fact that all the features are retained because the representation is passed directly to the next diffusion forward pass as compared to reasoning for discrete tokens which could only convey a specific type of representation if it is present in its vocabulary.

---

### Meta-Review · Area_Chair_rw6J · 2024-12-21

**Metareview:**

## Summary

This paper proposes a method based on a combination of latent diffusion models and encoder-decoder transformers for reasoning tasks. The authors argue that current autoregressive language models face fundamental limitations in reasoning, mainly due to their reliance on token-by-token processing. By leveraging latent diffusion, the proposed approach allows multiple iterative passes in latent space, enabling more flexible reasoning not constrained by token order. The model uses a fine-tuned encoder-decoder transformer for token-to-latent mapping and a diffusion process to predict target latent from input latent. Experimental results on reasoning benchmarks demonstrate that the latent diffusion approach is a scalable and effective alternative to autoregressive models for complex reasoning tasks.

## Decision

The paper is being rejected due to significant weaknesses raised by the reviewers, which point to major issues in methodology, clarity, experimental validation, and novelty.  Overall, the paper is poorly written and feels rushed. The reviewers identified shortcomings that make the contribution of the paper insufficient for acceptance at this conference. At the end of the rebuttal period the reviewers were not convinced with it and kept their original scores.

The primary reasons for rejection include unsupported claims, limited experimental scope, unclear comparisons, insufficient methodological details, and poor writing quality. Addressing these weaknesses in future revisions will be critical for the paper to be reconsidered.

**Additional Comments On Reviewer Discussion:**

1.	**Unsupported Claims and Lack of Evidence**

- Reviewer fxd5 noted that some claims made in the paper lack both qualitative and quantitative support.

- Reviewer BGPR also highlighted that many claims are not sufficiently supported by evidence or experiments.

- Reviewer Enpx mentioned that it is unclear how the proposed method differs from prior work and found the writing vague and lacking detail about key components.

	2.	**Limited Experiments and Unfair Comparisons**

- Reviewer BGPR criticized the experiments as being too simple, focusing on tasks like single-digit addition and spatial reasoning. They also pointed out unfair comparisons, such as evaluating the proposed method fine-tuned for tasks against GPT-3 in zero-shot settings.

- Reviewer NSA6 observed that only toy tasks were used, with no benchmarks like MMLU included. The results were not adequately contextualized or compared to other methods.

- Reviewer Enpx raised concerns about the lack of comparison with existing diffusion-based language models and questioned the fairness of evaluation setups.

	3.	**Insufficient Detail on Architecture and Methodology**

- Reviewer NSA6 and Reviewer Enpx both stated that the autoencoder and diffusion architecture internals were vague. Reviewer Enpx suggested adding diagrams or equations for clarity.

- Reviewer Enpx also emphasized the lack of explicit examples and details about tasks and ablation experiments, making it difficult to understand the contributions fully.

	4.	**Inapplicability to General Scenarios**
- Reviewer fxd5 questioned whether the method could scale to address the issues highlighted, especially in reasoning tasks like those on MMLU. They suggested narrowing the scope of the proposal.

- Reviewer Enpx noted that the fixed latent size imposes a hard limitation, making the method unsuitable for tasks requiring variable reasoning steps or higher complexity.

	5.	**Writing and Clarity Issues**
- Reviewer BGPR found many sentences and paragraphs unclear, providing examples of poorly written sections.
- Reviewer NSA6 also observed a lack of clarity in sections describing results and architecture.

---

### Decision · Program_Chairs · 2025-01-22

Reject